# Incorporation of Dietary Methyl Sulfonyl Methane into the Egg Albumens of Laying Hens

**DOI:** 10.3390/antiox11030517

**Published:** 2022-03-08

**Authors:** Yoo-Bhin Kim, Sang-Hyeok Lee, Da-Hye Kim, Hyun-Gwan Lee, Yong-Sung Jeon, Sung-Dae Lee, Kyung-Woo Lee

**Affiliations:** 1Department of Animal Science and Technology, Konkuk University, Gwangjin-gu, Seoul 05029, Korea; ybin51@naver.com (Y.-B.K.); the_shinism@naver.com (S.-H.L.); kdh142536@naver.com (D.-H.K.); leehyun3177@naver.com (H.-G.L.); fmdyd94@kakao.com (Y.-S.J.); 2National Institute of Animal Science, Rural Development of Administration (NIAS-RDA), Wanju 55365, Korea; leesd@korea.kr

**Keywords:** methyl sulfonyl methane, laying hen, antioxidant capacity, functional eggs

## Abstract

This study evaluated the effects of graded levels of dietary methyl sulfonyl methane (MSM) on the laying performance, egg quality, antioxidant capacity, and the incorporation of MSM into the egg albumen of laying hens. A total of 240 73-week-old laying hens (Lohmann Brown Lite) were randomly allotted to 1 of 5 dietary treatments, with 8 replicates of 6 birds per replicate. The experimental diets were formulated by mixing corn and soybean meal-based diets with MSM to reach 0.0, 1.0, 2.0, 3.0, and 4.0 g per kg of diet, and were fed to the birds for 12 weeks. Increasing dietary MSM led to a significant quadratic effect on the feed intake and feed conversion ratio at 4 weeks (*p* < 0.05). However, none of the egg qualities and egg components were altered by dietary MSM. The deposition of MSM in egg albumens increased in a linear manner (*p* < 0.05) in response to the increasing dietary MSM levels. The concentration of malondialdehyde in the egg yolk decreased at 12 weeks (linear and quadratic effect; *p* < 0.05), as the dietary MSM levels increased. Increasing dietary MSM affected the indicators of antioxidant/oxidative stress in the serum samples, such as superoxide dismutase at 12 weeks (linear and quadratic effect; *p* < 0.05), total antioxidant capacity at 8 and 12 weeks (linear effect; *p* < 0.05), and malondialdehyde at 8 weeks (linear effect; *p* < 0.05). Taken together, our study shows that dietary MSM has potential to be used as an antioxidant feed additive for laying hens, and can be used to produce functional eggs with health benefits for humans.

## 1. Introduction

Methyl sulfonyl methane (MSM) is a non-toxic natural organosulfur compound with the chemical formula (CH_3_)_2_SO_2_, and is known as methyl sulfone or dimethyl sulfone. It is a colorless solid with a sulfonyl functional group, and is the simplest form of sulfone. Due to its various biological properties, including antioxidant, antimicrobial, and anti-inflammatory, and its use for treating arthritis, MSM has been used as a functional supplement for human consumption [1,2]. MSM is naturally present in some natural green plants, fruits, and vegetables, such as tomato, corn, and apple [3,4], and is also present as a sulfur metabolite in animals [5,6]. MSM has been used in cosmetics for skin health [7], and in food supplements to alleviate joint and muscle pain [8,9], and is generally recognized as safe (GRAS) [10].

Due to the many biological advantages of MSM ingestion, dietary MSM in livestock production, including for chickens, has been increasingly investigated as an antioxidant [11,12], antimicrobial [13,14], and immune modulator [15]. As MSM is absorbed via the small intestine of animals, MSM-enriched animal products (e.g., milk) have been reported [16]. Earlier studies also reported that dietary MSM could enrich its concentration in the plasma, brains, and livers of broiler chickens [17,18], suggesting the possibility of effective transfer of dietary MSM into animal products. However, there are scarce data about the incorporation of dietary MSM into the eggs of laying hens, which prompted us to set up the current experiment. Based on the bioavailability evidence for dietary MSM [17,18], it was hypothesized that dietary MSM would be efficiently incorporated into the eggs in a dose-dependent manner. As dietary MSM acts as an antioxidant in poultry [11,12], we also assayed various markers of antioxidant capacity in the eggs and serum samples of laying hens. It is anticipated that the production of MSM-enriched eggs could be beneficial for humans (due to health benefits upon consumption) and egg producers ( who can charge a premium price).

## 2. Materials and Methods

### 2.1. Birds and Experimental Design

A total of 240 Lohmann Brown Lite laying hens, aged 73 weeks, were raised in a windowless chicken facility for 12 weeks. Two hens were raised per cage (45 cm × 45 cm) in a windowless, fan-ventilated house, and the adjacent 3 cages were considered a replicate (n = 6 birds/replicate, 8 replicates/treatment). After two weeks of an adaptation period, the layers were randomly distributed into five groups, with equal body weight and laying performance, and subjected to 1 of 5 experimental diets for 12 weeks. A corn and soybean meal-based diet was prepared (Table 1) to meet or exceed the nutritional requirements for brown egg-laying hens [19]. Experimental diets were formulated by adding MSM (Sigma Aldrich, St. Louis, MO, USA) into the basal diet to yield 1.0, 2.0, 3.0, and 4.0 g MSM per kg of diet. The diet with no MSM added was used as the control diet.

The MSM content was analyzed for the control diet and it contained 0.31 g/kg of diet. MSM is a natural constituent of the environment and it is found in plants, including corn [20]. Total S contents were analyzed with an elemental analyzer (EA 1110 CHN; CE instruments, Rodano, MI, Italy) and the control diet was found to contain 1.5 g S/kg of diet. Lim et al. [21] suggested that dietary MSM can be used from 0.1 to 4.0 g per kg of diet. In addition, the toxic level was defined as 4.85 g/day [22]. Thus, we decided to gradually add MSM up to 4.0 g per kg of diet. Feed and water were supplied to allow ad libitum consumption during the experimental period. A lighting program of 15 h of light and 9 h of dark was used for the entire experimental period. The temperature and relative humidity of the experimental room were maintained at 21 ± 2 °C and 60%, respectively.

### 2.2. Laying Performance Parameters and Egg Quality

The feed consumption per replicate was recorded monthly and used to calculate the average daily feed intake per bird. Feed was provided daily to avoid the chances of feed wastage. Egg production and egg weight were recorded daily and used to calculate the egg mass. The percentage of dirty and broken eggs was calculated as (total number of dirty and broken eggs per replicate/total number of eggs per replicate) × 100. The feed conversion ratio (FCR; feed/egg mass) was calculated on a 4-week basis for each group. 

On the last three consecutive days at 4, 8, and 12 weeks, six intact eggs per replicate were collected for the egg quality assessment. Eggshell color was estimated by a shell color reflectometer (TSS QCR, Technical Services and Supplies, York, UK). Haugh unit, eggshell strength, eggshell thickness (without shell membrane) and yolk color score were assessed by a digital egg tester (DET6000, Nabel, Kyoto, Japan). Yolk color was automatically graded on a scale of 1–16, with 1 being a very pale yellow and 16 being a dark orange. The separated yolks were weighed after clearing the adherent albumen residue with filter paper. Eggshells were cleaned to remove the adherent albumen, dried at room temperature for 3 days, and weighed. Albumen weight was then calculated by subtracting yolk and dry eggshell weights from the initial whole egg weight. The separated yolks were pooled (3 yolks/replicate) and processed to determine the concentrations of corticosterone and malondialdhyde (MDA), as described elsewhere [23,24,25].

### 2.3. MSM Content in Egg Albumen

In a preliminary analysis, we found that egg yolks contained no MSM, but the albumen contained 0.18 g/kg (data not shown). Thus, we decided to analyze the albumen for MSM incorporation. At 12 weeks, three eggs per replicate were collected to measure the MSM content in egg albumen. The eggs were cracked open and yolk-free albumen was separated and pooled. The pooled albumen per replicate was homogenized and lyophilized (Labtech Freeze Dryer, Daihan, Korea). After lyophilization, the dried albumen was ground, dissolved and extracted in 50 mL methanol with an internal standard (diethylene glycol methyl ether, Sigma Aldrich, St. Louis, MO, USA) in a 100 mL volumetric flask [26]. Then, the extracts were filtered through a 0.45 μm filter (Whatman 6789-1304 syringe filter, Piscataway, NJ, USA), and 1 μL of the filtrate was injected directly into a gas chromatograph (HP 6890 series GC system, Agilent Tech, CA, USA) equipped with a DB-1 capillary column (0.32 mm I.D. × 30 m, 0.25 μm, dimethylpolysiloxane) and a flame ionization detector. Nitrogen was used as the carrier gas at a flow rate of 1.0 mL/min. The detection limit for MSM by gas chromatography was estimated to be 0.29 μg/g.

### 2.4. Corticosterone and Malondialdehyde in Egg Yolk

Malondialdehyde is an indicator of oxidative stress [24,27], and corticosterone is a marker for stress [23] in chickens. We attempted to analyze the concentrations of corticosterone and MDA in egg yolks to evaluate the role of MSM in the stress response. Pooled egg yolks were used to measure corticosterone (Enzo life science Inc., ADI-901-097, Farmingdale, NY, USA) and MDA (Cell Biolabs, Inc., San Diego, CA, USA), as per the recommendations of the manufacturers. 

### 2.5. Antioxidant Markers in Serum Samples

Approximately 3 mL of blood per hen (one hen per replicate) was drawn from the wing vein into a clot activator tube (Becton Dickinson Vacutainer, Franklin Lakes, NJ, USA) at 4, 8, and 12 weeks. Serum samples were obtained by gentle centrifugation (200× *g*) for 15 min and stored at −20 °C before analysis. Serum samples were used to measure various biomarkers of oxidative stress, including levels of glutathione peroxidase (GPX), superoxide dismutase (SOD), total antioxidant capacity (TAC), catalase (CAT), MDA, and 8-hydroxydeoxyguanosine (8-OHdG). The level of activity of GPX in serum samples was determined using a GSH-Px ELISA kit (EnzyChrom GPx, BioAssay Systems, Hayward, CA, USA). SOD was analyzed using a SOD determination assay kit (Sigma, St. Louis, MO, USA) and expressed as SOD activity (% inhibition rate). TAC was analyzed using a QuantiChrom antioxidant assay kit (BioAssay Systems, Hayward, CA, USA) and expressed as mmol Trolox equivalents. CAT was analyzed using an OxiSelect catalase activity assay kit (Cell Biolabs, Inc., San Diego, CA, USA). MDA was measured using an OxiSelect thiobarbituric acid reactive substances (TBARS) assay kit (Cell Biolabs, Inc., San Diego, CA, USA). 8-OHdG was determined based on oxidative DNA damage (Cell Biolabs, Inc., San Diego, CA, USA) and expressed in ng/mL.

### 2.6. Statistical Analysis

Three adjacent cages were considered as the experimental unit. All data were analyzed using the general linear model procedures of SAS (version 9.4; SAS Institute Inc., Cary, NC, USA). Orthogonal polynomial contrasts were used to determine the linear and quadratic effects of graded levels of dietary MSM. The significance level was set at *p* < 0.05.

## 3. Results

### 3.1. Laying Performance and Egg Quality

The production performance of the laying hens fed diets with graded levels of dietary MSM is presented in Table 2. At 4 weeks, increasing dietary MSM quadratically decreased the feed intake (*p* = 0.011) and FCR (*p* = 0.049). However, the MSM-mediated effect on feed intake and FCR was not detected at 8 and 12 weeks. No significant (*p* > 0.05) linear and quadratic effects of dietary MSM on egg production, egg weight, and egg mass at all timepoints were noted. Dirty and cracked egg production remained between 1.11 and 2.98%, and was not affected by dietary MSM levels. Dietary MSM failed to affect the egg components and egg qualities, including the Haugh unit, eggshell thickness, or eggshell strength, at all timepoints (Table 3). 

### 3.2. MSM Deposition in Egg Albumen

The MSM content in the egg albumen increased (linear; *p* < 0.01) as the dietary MSM levels increased (Figure 1). Of interest, MSM was detected at 0.6 g per kg of albumen in the control group. Increasing the dietary MSM levels resulted in a linear equation of Y = 9.4319X + 0.6041 (R^2^ = 0.963, where Y presents the MSM content (g/kg) in the egg albumen, and X represents the level of added MSM (%) in the diets).

### 3.3. Corticosterone and MDA in Yolk Samples

Dietary MSM supplementation did not affect the concentration of corticosterone in the egg yolks at all timepoints (*p* > 0.05; Table 4). As an indicator of oxidative stress, the MDA content of the egg yolks decreased linearly and quadratically (*p* < 0.05) with increasing dietary MSM levels, but this effect was only noted at 12 weeks.

### 3.4. Markers for Antioxidant and Oxidative Stress in Serum Samples

The serum concentrations of the antioxidant/oxidative stress markers, including GPX, SOD, TAC, CAT, MDA, and 8-OHdG, are presented in Table 5. Dietary MSM did not affect GPX, CAT and 8-OHdG in the serum samples at any time. At 12 weeks, the SOD activity in the serum samples was elevated (linear and quadratic effects; *p* < 0.05) by increasing the dietary MSM levels. Dietary MSM did not affect the concentration of TAC in the serum samples at 4 weeks, but it had increased linearly (*p* < 0.05) by 8 and 12 weeks. The MDA concentrations decreased in the serum samples (*p* > 0.05) with increasing dietary MSM levels, and this effect was significant (linear effect; *p* = 0.025) at 8 weeks and marginal (linear effect; *p* = 0.085) at 12 weeks.

## 4. Discussion

It is clear from this study that dietary MSM can be linearly and effectively transferred into the egg albumen. However, despite supplementary MSM levels of up to 4.0 g per kg of diet, no plateau of MSM incorporation was reached for the egg albumen. In line with our finding, the systemic incorporation of dietary MSM has been reported in broiler chickens. For example, Abdul Rasheed et al. [17] reported that intact MSM was easily detectable in the plasma and tissues (i.e., liver, spleen, heart, kidney, brain, cecal tonsil, hock joint, and abdominal skin) of broiler chickens fed 1.0 or 2.0 g MSM/kg of diet. Furthermore, dietary MSM at 0.3 g per kg of diet is known to increase the sulfur content of duck breast meat [28]. Lim et al. [21] reported that the supplementation of MSM at the concentrations of 1.0, 2.0, and 4.0 g/kg of diet significantly increased the total sulfur concentration in the eggs. Thus, the MSM-mediated increase in the sulfur content of eggs and breast meats [21,28] originated from dietary MSM that had been absorbed and deposited systemically. It is well known that dietary sulfur per se is not used to synthesize sulfur-containing amino acids (i.e., methionine and cysteine) in chickens. Indeed, Shin et al. [29] reported that feeding sulfur at 0.2% in the diet did not affect the sulfur-containing amino acids in chicken meat. As far as we know, this is the first report to describe the transfer of dietary MSM into chicken eggs, and to determine a nutritional intervention to develop MSM-enriched eggs. However, future studies are needed to determine the dietary levels that can maximize the incorporation of MSM into the egg albumen. 

It has been well reported that dietary MSM plays an important role in reducing oxidative stress [12,15], inhibiting pathogenic bacteria [30], and improving the host’s immune response [31]. In addition, MSM is known to inhibit the cortisol-induced stress response in horse skeletal muscle cells. Thus, we attempted to measure various biological markers, including the general stress response and oxidative stress. It is known that the level of corticosterone in the egg albumen increases in laying hens exposed to stressors [32]. However, we did not find any effect of dietary MSM on egg yolk corticosterone, indicating a negligible effect on the stress physiology of laying hens. 

It is known that about 99% of all egg lipids are present in the yolk, with a large percentage of unsaturated fatty acids prone to high lipid oxidation [33]. In this study, the graded levels of dietary MSM linearly and quadratically lowered the concentration of MDA in the egg yolks at 12 weeks, confirming its antioxidative activity. However, it is not clear, at this stage, why dietary MSM did not affect the concentration of MDA in the eggs sampled at 4 and 8 weeks. Analyzing the MSM content of the egg yolks would, in part, provide an explanation for the observed delay of MSM in reducing the MDA concentration.

We further monitored the antioxidant properties of serum samples from laying hens fed a diet containing dietary MSM. Dietary MSM increased SOD and TAC, but lowered MDA, in the serum samples. It is of interest that the oxidative activity of dietary MSM was noted at 8 or 12 weeks, indicating a time-dependent antioxidant effect of MSM. In line with our finding, 0.03% dietary MSM markedly increased the SOD in ducklings [28]. Similarly, Pekin ducks fed a diet supplemented with 0.3% MSM had higher serum activities for SOD and TAC, but lower serum MDA concentrations, when compared with the control group [11]. 

In regards to laying performance, increasing the dietary inclusion of MSM quadratically decreased feed intake and improved FCR, although this effect was only detected at 4 weeks. Our study, in part, corroborates earlier studies that showed that dietary MSM decreased feed intake in ducks [11] and improved FCR in broiler chickens [14]. However, whether the MSM-mediated improvement in FCR in this study is linked to an increase in nutrient utilization or the activity of digestive enzymes awaits to be addressed. In contrast to our findings, dietary MSM did not affect the production performance of laying hens [21,34], or the growth performance of ducklings [28] and broiler chickens [17]. Thus, the effect of MSM on the production performance of chickens is, at best, inconclusive. 

Although none of the egg qualities were affected by dietary MSM in this study, Lim et al. [21] reported that the supplementation of 1.0, 2.0, and 4.0 g MSM per kg of diet increased the albumen height and Haugh unit. In addition, significant increases in the Haugh unit and heights of the egg yolk and albumen were noted in dietary MSM-fed laying hens [34]. Whether the discrepancies between our study and others [21,34] might be linked to the age of the birds needs to be addressed. We used birds aged 73 weeks, while the birds were 35 weeks of age in the study by Park et al. [34] and 31 weeks of age in the study by Lim et al. [21]. 

## 5. Conclusions

In conclusion, graded levels of dietary MSM linearly improved the antioxidant capacity in laying hens, as manifested by a decrease in MDA in the egg yolks (12 weeks), and an increase in SOD (12 weeks) and TAC (8 and 12 weeks) in the serum samples. However, it should be remembered that the MSM-mediated effects on the antioxidant and oxidative markers were exhibited in a time-dependent manner. In addition, we found that increasing MSM in the diets of laying hens linearly increased the deposition of MSM in the egg albumen. Our study shows that dietary MSM could be used as a functional additive to produce value-added MSM-enriched eggs.

## Figures and Tables

**Figure 1 antioxidants-11-00517-f001:**
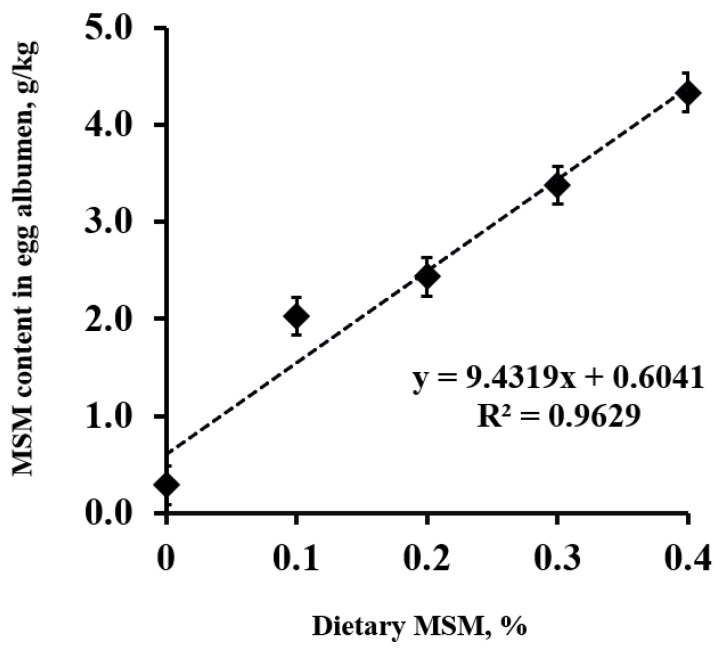
Methyl sulfonyl methane (MSM) content (g/kg) in lyophilized egg albumen from laying hens fed a control diet or an MSM-enriched diet containing 1.0, 2.0, 3.0, and 4.0 g MSM per kg of diet.

**Table 1 antioxidants-11-00517-t001:** Ingredient and nutrient composition of the basal diet.

Ingredients	g per 100 g of Diet
Corn	42.59
Soybean meal, 45% crude protein	10.41
Wheat	12.80
Animal fat	1.02
Rice bran	2.00
Corn steep liquor	1.00
Rapeseed meal	3.00
Dried distillers grains with solubles	12.83
Molasses	2.00
Liquid choline, 50%	0.06
Limestone	10.51
Monodicalcium phosphate	1.02
NaCl	0.24
Methionine, 99%	0.07
Lysine sulfate, 54%	0.10
Tryptophane, 10%	0.10
Mineral mix ^1^	0.14
Vitamin mix ^2^	0.12
Total	100.00
Nutrient composition, g/100 g	
Nitrogen-corrected apparent metabolizable energy ^3^, kcal/kg	2600
Dry matter ^4^	88.20
Crude protein ^4^	14.49
Crude fat ^4^	4.01
Crude ash ^4^	14.84
Calcium ^3^	4.10
Sulfur ^4^	0.15
Available phosphorus ^3^	0.28
Total lysine ^3^	0.65
Total methionine ^3^	0.32
Total methionine + cysteine ^3^	0.60
Methyl sulfonyl methane ^4^	0.03

^1^ Mineral mixture provided the following nutrients per kg of diet: Fe (FeSO_4_.H_2_O), 50 mg; Cu (CuSO_4_.H_2_O), 24 mg; Zn (ZnSO_4_.H_2_O), 90 mg; Mn (MnSO_4_.H_2_O), 96 mg; I (Ca(IO_3_)_2_), 1.2 mg. ^2^ Vitamin mixture provided the following nutrients per kg of diet: vitamin A, 15,400 IU; vitamin D_3_, 3080 IU; vitamin E, 14 mg; vitamin K_3_, 1.4 mg; vitamin B_1_, 1.12 mg; vitamin B_2_, 2.8 mg; vitamin B_6_, 3.92 mg; vitamin B_12_, 0.014 mg. ^3^ Calculated value. ^4^ Analyzed value.

**Table 2 antioxidants-11-00517-t002:** Effects of graded levels of dietary methyl sulfonyl methane on the laying performance of laying hens.

Item	Dietary MSM, %					SEM ^1^	*p*-Value	
	0	0.1	0.2	0.3	0.4		Linear	Quadratic
Feed intake, g/bird								
4 weeks	103.59	99.67	101.13	101.23	103.63	0.99	0.656	0.011
8 weeks	100.07	101.19	103.26	101.69	103.79	1.89	0.280	0.839
12 weeks	107.58	111.05	107.93	105.89	112.02	1.98	0.587	0.422
Egg production, %								
4 weeks	72.62	79.17	79.91	78.63	79.37	2.97	0.249	0.284
8 weeks	77.09	81.03	82.29	80.15	83.59	3.45	0.365	0.769
12 weeks	77.30	80.56	79.63	77.25	85.37	2.61	0.163	0.432
Egg weight, g								
4 weeks	62.56	63.87	62.00	62.67	61.52	0.78	0.263	0.471
8 weeks	63.76	63.73	63.70	64.05	62.02	0.66	0.217	0.212
12 weeks	64.08	63.84	64.44	65.09	63.92	0.70	0.702	0.523
Egg mass, g/day								
4 weeks	45.36	50.53	49.50	49.24	48.86	1.83	0.407	0.191
8 weeks	49.11	51.48	52.43	51.32	51.85	2.12	0.518	0.535
12 weeks	49.38	51.37	51.33	50.32	54.66	1.73	0.121	0.593
Feed conversion ratio, kg/kg								
4 weeks	2.38	1.98	2.06	2.06	2.14	0.10	0.297	0.049
8 weeks	2.12	1.98	1.97	2.00	2.01	0.09	0.587	0.412
12 weeks	2.19	2.18	2.11	2.11	2.07	0.06	0.152	0.972
Dirty and cracked egg, %								
4 weeks	2.73	2.72	2.63	2.44	2.33	0.68	0.644	0.911
8 weeks	2.98	2.82	2.32	1.11	1.87	0.79	0.132	0.706
12 weeks	2.84	2.09	2.27	2.24	2.06	0.75	0.609	0.773

^1^ SEM, standard error of the mean.

**Table 3 antioxidants-11-00517-t003:** Effects of graded levels of dietary methyl sulfonyl methane on the egg components and egg quality from laying hens.

Item	Dietary MSM, %					SEM ^1^	*p*-Value	
	0	0.1	0.2	0.3	0.4		Linear	Quadratic
4 weeks								
Relative yolk weight, %	26.02	26.06	25.50	27.34	25.97	0.80	0.645	0.887
Relative eggshell weight, %	9.95	9.47	10.14	9.79	9.73	0.27	0.905	0.862
Relative albumen weight, %	64.12	64.54	64.47	62.87	64.35	0.89	0.673	0.860
Yolk color	6.01	6.05	6.05	6.08	6.00	0.07	0.980	0.473
Haugh unit	73.85	71.35	74.22	72.66	73.25	1.59	0.983	0.787
Eggshell strength, kg/cm^2^	4.80	4.77	4.73	4.64	4.48	0.17	0.169	0.627
Eggshell thickness, mm	0.42	0.42	0.43	0.42	0.42	0.01	0.971	0.384
Eggshell color, unit	28.24	28.01	27.83	30.46	27.66	0.80	0.610	0.442
8 weeks								
Relative yolk weight, %	26.67	26.49	26.01	26.86	26.60	0.36	0.833	0.393
Relative eggshell weight, %	10.05	10.12	10.00	10.03	9.98	0.12	0.540	0.851
Relative albumen weight, %	63.49	63.40	63.98	63.12	63.39	0.35	0.662	0.583
Yolk color	6.76	6.70	6.62	6.75	6.65	0.08	0.511	0.640
Haugh unit	75.89	76.22	76.09	74.91	76.18	1.05	0.832	0.837
Eggshell strength, kg/cm^2^	4.61	4.44	4.63	4.77	4.76	0.18	0.274	0.671
Eggshell thickness, mm	0.41	0.41	0.41	0.40	0.40	0.01	0.150	0.932
Eggshell color, unit	27.48	27.50	28.01	28.01	27.97	0.54	0.454	0.773
12 weeks								
Relative yolk weight, %	25.72	25.73	25.40	27.24	25.74	0.57	0.395	0.688
Relative eggshell weight, %	9.96	9.40	9.92	9.81	9.67	0.22	0.827	0.787
Relative albumen weight, %	64.32	64.93	64.67	62.94	64.66	0.58	0.477	0.738
Yolk color	6.70	6.67	6.71	6.70	6.70	0.07	0.951	0.953
Haugh unit	75.86	75.98	77.05	75.60	79.05	1.38	0.190	0.438
Eggshell strength, kg/cm^2^	4.51	3.97	4.50	4.45	4.23	0.17	0.865	0.942
Eggshell thickness, mm	0.41	0.42	0.40	0.40	0.40	0.01	0.207	0.901
Eggshell color, unit	9.96	9.40	9.92	9.81	9.67	0.22	0.827	0.787

^1^ SEM, standard error of the mean.

**Table 4 antioxidants-11-00517-t004:** Effects of graded levels of dietary methyl sulfonyl methane on the levels of corticosterone and malondialdehyde in the egg yolks from laying hens.

Item ^1^	Dietary MSM, %					SEM ^2^	*p*-Value	
	0	0.1	0.2	0.3	0.4		Linear	Quadratic
Corticosterone, pg/g								
4 weeks	875.27	807.22	803.05	825.92	849.40	139.00	0.962	0.785
8 weeks	270.12	258.12	245.87	222.10	244.53	21.67	0.307	0.548
12 weeks	407.04	392.19	395.68	375.02	383.25	32.30	0.565	0.865
MDA, nmol/g								
4 weeks	36.50	36.58	35.08	36.33	37.08	1.53	0.874	0.542
8 weeks	28.82	29.44	28.93	28.95	28.43	1.30	0.777	0.735
12 weeks	28.01	22.36	21.77	21.95	21.24	1.06	<0.001	0.012

^1^ MDA, malondialdehyde. ^2^ SEM, standard error of the mean.

**Table 5 antioxidants-11-00517-t005:** Effects of graded levels of dietary methyl sulfonyl methane on the serum markers of oxidative stress in laying hens.

Item ^1^	Dietary MSM, %					SEM ^2^	*p*-Value	
	0	0.1	0.2	0.3	0.4		Linear	Quadratic
GPX activity, U/L								
4 weeks	422.39	443.11	448.94	442.97	439.11	57.80	0.858	0.781
8 weeks	426.95	440.61	435.6	448.75	423.32	55.90	0.997	0.812
12 weeks	458.01	463.67	489.30	489.87	487.87	37.95	0.527	0.765
SOD activity, %								
4 weeks	74.07	77.53	80.77	81.07	89.16	7.65	0.257	0.854
8 weeks	74.88	77.05	79.57	78.67	77.65	6.03	0.766	0.741
12 weeks	77.29	98.39	103.30	108.53	106.50	5.01	0.002	0.044
TAC, mM								
4 weeks	1.21	1.45	1.34	1.36	1.30	0.20	0.893	0.577
8 weeks	1.04	1.15	1.29	1.22	1.36	0.09	0.018	0.652
12 weeks	1.11	1.14	1.32	1.55	1.55	0.13	0.014	0.995
CAT, U/mL								
4 weeks	3.06	2.89	2.41	2.90	2.87	0.29	0.761	0.347
8 weeks	2.60	3.68	3.46	3.10	3.47	0.70	0.661	0.607
12 weeks	2.25	2.30	2.52	2.51	2.86	0.21	0.203	0.742
MDA, µM								
4 weeks	26.05	20.02	20.08	23.02	21.75	3.76	0.676	0.424
8 weeks	25.97	26.61	25.40	20.34	19.12	2.19	0.025	0.430
12 weeks	26.30	23.02	21.93	19.87	20.59	2.13	0.085	0.456
8-OHdG, ng/mL								
4 weeks	1.74	1.37	1.62	1.85	1.64	0.25	0.755	0.752
8 weeks	2.92	3.63	3.24	3.31	3.31	0.32	0.735	0.530
12 weeks	2.09	1.51	1.56	1.52	1.32	0.34	0.260	0.694

^1^ GPX, glutathione peroxidase; SOD, superoxide dismutase; TAC, total antioxidant capacity; CAT, catalase; MDA, malondialdehyde; 8-OHdG, 8-hydroxydeoxyguanosine. ^2^ SEM, standard error of the mean.

## Data Availability

Data is contained within the article.

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
