# Peer review of "Incorporation of Dietary Methyl Sulfonyl Methane into the Egg Albumens of Laying Hens"

_antioxidants, 2022, doi:10.3390/antiox11030517_

Round 1

Reviewer 1 Report

The research explores  the effect of supplementing MSM on its incorporation  into the Egg Albumens of Laying Hens

In general terms it is a good paper, only minor comments (see file) which I believe author can solve withour problem

My main recommendation is related to the need to present a clear mechanistic hypothesis and use it to guide the discussion. This will remark the contribution of the present paper to the knowledge in this area.

currently author mostly present the need for further research but fall short into providing possible explanation of the results, or lack of, base on the hypotesis.

Author Response

Reviewer #1:

The research explores the effect of supplementing MSM on its incorporation into the Egg Albumens of Laying Hens. In general terms it is a good paper, only minor comments (see file) which I believe author can solve withour problem. My main recommendation is related to the need to present a clear mechanistic hypothesis and use it to guide the discussion. This will remark the contribution of the present paper to the knowledge in this area. currently author mostly present the need for further research but fall short into providing possible explanation of the results, or lack of, base on the hypotesis.

Authors’ Response: We appreciate the Reviewer’s comments which helped us to significantly improve the manuscript.

Line 47-48: it is suggested to include an hypothesis. The hypothesis allows to properly evaluate what authors want to test and the appropriateness of experimental design. After reading the discussion, it is clear that it needs an hypothesis guiding the discussion.

Authors’ Response: Per the Reviewer’s suggestion, we have now added hypothesis (L49-51):

“Based on the bioavailable evidence of dietary MSM [17,18], it can be hypothesized that dietary MSM would be efficiently incorporated into the eggs in a dose dependent manner.”

Line 57:

Authors’ Response: Per the Reviewer’s comment, we have now corrected to “per” (L59).

Line 78: how? was feed wastage (fell to the floor) inlcuded or corrected for? how or why not?

Authors’ Response: We have addressed how to prevent feed wastage as follows (L81-82).

“Feed was provided daily to avoid the chances of feed wastage.”

Line 85: can MSM affect shell color?

Authors’ Response: We decided to analyze the eggshell color as it is considered one of egg quality parameters.

Line 153: Generally orthogonal polynomial contrasts (OPC) are correcto for treatment which do have level/dosage effects. The multiple range could be not required as the effect must be explained in terms of the singnificance of the OPC. For example, if there is a significant linear effect, then, it is not need to compare individual values as the effect is already declared and the change (increase or decrease) in any particular value it is assumed significant as describe by the equation describing the relationship.

after reading the paper, the duncan comparisom can be deleted as the discussion is properly managed with reference to OPC and not duncan test.

Authors’ Response: We agree with the Reviewer’s comments. Per the Reviewer’s recommendation, we have now deleted Duncan test in the text and P values in Tables.

Reviewer 2 Report

Authors have prepared an interesting word. However, it is necessary to have some important corrections before final evaluation.

2.4 corticosterone and MDA in egg yolk have been meaasured by assay kits, this is not the most accurate way of measurement.

Also, serum samples have analysed by kits, are those the same?

Haugh units are relatively low and show that eggs were not fully fresh. Do they have an explanation

MDA was found different only after 12 weeks. This difference was not noticed in serum.

By those results, conclusions are not concrete.

line 280, in MDA only after 12 weeks, 

lines 282-284, this porposal in not stronlgy supported by the data, and its more a wish, most practical would be that MSN does not produce lipid oxidation??? 

Author Response

Authors have prepared an interesting word. However, it is necessary to have some important corrections before final evaluation.

2.4 corticosterone and MDA in egg yolk have been measured by assay kits, this is not the most accurate way of measurement. Also, serum samples have analysed by kits, are those the same?

Authors’ Response: We agree with the Reviewer’s comment that ELISA for corticosterone and MDA in egg yolk is not the most accurate way of measurement. However, we all agree that ELISA-based assays for corticosterone and MDA in egg yolk samples have been widely used in many laboratories due to the technical advantages including simplicity, reliability and high-throughput implementation with accuracy and specificity. In addition, we have used the same kits for both egg yolks and serum samples.

We have now added the references that used the same ELISA kits to measure corticosterone and MDA in egg yolks and serum samples.

Kim, D.H.; Lee, Y.K.; Kim, S.H.; Lee, K.W. The impact of temperature and humidity on the performance and physiology of laying hens. Animals 2021, 11, 1–12.

Lee, S.H.; Kim, Y.B.; Kim, D.-H.; Lee, D.-W.; Lee, H.-G.; Jha, R.; Lee, K.-W. Dietary soluble flaxseed oils as a source of omega-3 polyunsaturated fatty acids for laying hens. Poult. Sci. 2021, 100, 101276.

Kim, Y.; Lee, S.; Kim, D.; Lee, H.; Choi, Y.; Lee, S.; Lee, K. Effects of dietary organic and inorganic sulfur on laying performance, egg quality, ileal morphology, and antioxidant capacity in laying hens. Animals 2022, 12, 1–12.

Haugh units are relatively low and show that eggs were not fully fresh. Do they have an explanation

Authors’ Response: It is well understood that egg quality including Haugh unit declines with increasing hen age. As indicated in Materials and Method, we have used aged 73-week-old laying hens producing relatively poorer egg quality compared with the younger hens.

MDA was found different only after 12 weeks. This difference was not noticed in serum. By those results, conclusions are not concrete.

Authors’ Response: Per the Reviewer’s comment, we have rephrased the conclusion to indicate time-dependent effects on the antioxidant and oxidative markers as follows (L277-279).

“However, it should be remembered that the MSM-mediated effects on the antioxidant and oxidative markers exhibited a time-dependent manner.”

Line 280: in MDA only after 12 weeks, 

Authors’ Response: Per the Reviewer’s comment, we have now added as follows (L275-277).

“In conclusion, graded levels of dietary MSM linearly improved the antioxidant ca-pacity in laying hens as manifested by a decrease in MDA in egg yolks (12 weeks) and an increase in SOD (12 weeks) and TAC (8 and 12 weeks) in serums samples.”

Lines 282-284: this porposal in not stronlgy supported by the data, and its more a wish, most practical would be that MSN does not produce lipid oxidation??? 

Authors’ Response: Per the Reviewer’s comment, we have now deleted “with improved antioxidant capacity” from the sentence.

Reviewer 3 Report

Line 60 change “six groups” as “five groups”

Table 1: Variable 1.59 what ingredient is ?

Line 119  after oxidative stress [24] please add reference  (Kalvandi, O.; Sadeghi, A.; Karimi, A. Arginine supplementation improves reproductive performance, antioxidant status, immunity and maternal antibody transmission in breeder Japanese quail under heat stress conditions. Ital. J. Anim. Sci. 2022, 21, 8-17. https://doi.org/10.1080/1828051X.2021.2013136)

Line 206 change “p < 0.05)” as “p > 0.05)”

Line 341 change “Richmond, V.L. Incorporation of methylsulfonylmethane sulfur info guinea pig serum proteins. 1986, 39, 263–268.” as “Richmond, V.L. Incorporation of methylsulfonylmethane sulfur info guinea pig serum proteins. Life Sci. 1986, 39, 263–268.”

Author Response

Line 60: change “six groups” as “five groups”

Authors’ Response: Per the Reviewer’s comment, we have now corrected to “five groups” (L62).

Table 1: Variable 1.59 what ingredient is ?

Authors’ Response: It is corn. We have now corrected it in Table 1.

Line 119: after oxidative stress [24] please add reference  (Kalvandi, O.; Sadeghi, A.; Karimi, A. Arginine supplementation improves reproductive performance, antioxidant status, immunity and maternal antibody transmission in breeder Japanese quail under heat stress conditions. Ital. J. Anim. Sci. 2022, 21, 8-17. https://doi.org/10.1080/1828051X.2021.2013136)

Authors’ Response: Per the Reviewer’s suggestion, we have now added the reference.

Line 206: change “p < 0.05)” as “p > 0.05)”

Authors’ Response: Per the Reviewer’s comment, we have now corrected to “p > 0.05” (L206).

Line 341: change “Richmond, V.L. Incorporation of methylsulfonylmethane sulfur info guinea pig serum proteins. 1986, 39, 263–268.” as “Richmond, V.L. Incorporation of methylsulfonylmethane sulfur info guinea pig serum proteins. Life Sci. 1986, 39, 263–268.”

Authors’ Response: Per the Reviewer’s comment, we have now corrected (L339).

Round 2

Reviewer 2 Report

Authors have revised their work in a very satisfactory way.